

# Correlation between pre-operative VE-cadherin and DLL4 and the maturation after primary arteriovenous fistula in uremic patients

Huanhuan Yin[1], Yifan Tang[1], Yanping Wang[1],
Yousuf Abdulkarim Waheed[1], Disheng Wang[1] and Dong Sun[1,2,3]

[1] Department of Nephrology, Affiliated Hospital of Xuzhou Medical University, Xuzhou, China
[2] Department of Internal Medicine and Diagnostics, Xuzhou Medical College, Xuzhou, China
[3] Clinical Research Center for Kidney Disease, Xuzhou Medical University, Xuzhou, Jiangsu, China

## ABSTRACT

**Aims:** Uremic patients require dialysis to replace the declined kidney function, and arteriovenous fistula (AVF) is a commonly used dialysis access route. Our study aimed to explore vascular endothelial cells cadherin (VE-cadherin) and Delta-like ligand 4 (DLL4) expression in uremic patients undergoing primary AVF surgery and their correlation with AVF maturation.

**Methods:** We conducted a prospective study that included $n = 55$ voluntary uremic patients receiving their initial AVF procedure for renal replacement therapy, subjects were divided into a mature group and a failure group based on whether the AVF matured within 3 months post-operatively. We analyzed the association of VE-cadherin and DLL4 with AVF maturation by examining their expression levels in serum and the endothelium of cephalic veins.

**Results:** Pre-operative serum VE-cadherin, in the mature group measured 125.07 (106.77–167.65) ng/L, and DLL4 was 92.78 (83.83–106.72) pg/mL, while the failure group had VE-cadherin at 95.40 (79.03–107.16) ng/L ($P = 0.001$), and DLL4 at 60.42 (43.98–80.15) pg/mL with a statistical significant; ($P = 0.002$), binary logistic regression analysis indicated a significant association between cephalic vein diameter, VE-cadherin, DLL4 levels, and AVF immaturity ($P = 0.024$, $P = 0.014$ respectively). Immunohistochemical staining showed slightly higher VE-cadherin levels in the mature group than in the failure group ($P = 0.366$). DLL4 was primarily located in the cell membrane and cytoplasm, concentrated in the inner membrane, with significantly higher levels in the mature group compared to the failure group ($P = 0.027$).

**Conclusion:** The failure group exhibited lower levels of VE-cadherin and DLL4 in serum and vascular tissue, these results suggest that VE-cadherin and DLL4 might play pivotal regulatory roles in the onset and the progression of fistula immaturity, potentially serving as promising targets for future interventions.

Corresponding author
Dong Sun, sundongxz@126.com

## INTRODUCTION

Patients with end-stage renal disease (ESRD) rely on hemodialysis to replace their deteriorated kidney function, with the arteriovenous fistula (AVF) being a commonly used access route for hemodialysis. Although the AVF is considered the preferred long-term vascular access for hemodialysis due to its durability and lower risk of infection and thrombosis (*Murea, Grey & Lok, 2021*; *Lok et al., 2020*), the failure of AVF maturation remains a significant challenge for both patients and healthcare providers. Failure to mature can lead to the loss of vascular access, affecting the effectiveness of dialysis treatment. Therefore, preventing AVF loss and maintaining access patency are crucial in the management of ESRD hemodialysis.

The effectiveness of hemodialysis treatment depends on the proper functioning of vascular access. Successful AVF creation relies on vascular maturation, in which vascular endothelial cells play a crucial role. They facilitate vascular connectivity, enhance stability, and ensure proper function (*Krüger-Genge et al., 2019*). AVF maturation integrates outward remodeling with vessel wall thickening in response to drastic hemodynamic changes, uremia, systemic inflammation, oxidative stress, and preexistent vascular pathology. AVF non-maturation is often caused by a narrowing near the connection point, known as juxta-anastomotic stenosis, as well as aggressive neointimal hyperplasia at the juxta-anastomosis on a histological level and a lack of outward vascular remodeling (*Siddiqui et al., 2017*). According to *Eloueyk et al. (2019)* uremic environment induces cell apoptosis and hinders cell proliferation, leading to endothelial dysfunction, which contributes to abnormal vascular remodeling and neointimal hyperplasia (NIH) at the anastomosis of vascular access, ultimately resulting in vascular access failure. Failure due to NIH involves vascular cell activation and migration and extracellular matrix remodeling with complex interactions of growth factors, adhesion molecules, inflammatory mediators, and chemokines, all of which result in maladaptive remodeling (*Hu et al., 2016*).

Vascular endothelial cells cadherin (VE-cadherin) and Delta-like ligand 4 (DLL4) have gained significant attention as key factors governing endothelial cell interactions and vascular stability. VE-cadherin, part of the classical cadherin superfamily, is specifically expressed in vascular endothelial cells (*Gavard, 2014*). VE-cadherin plays a pivotal role in mediating the stabilization of endothelial cell (EC) junctions, contributing to EC migration, survival, contact-induced growth inhibition, regulation of vascular remodeling, and the preservation of vascular integrity (*Lampugnani, Dejana & Giampietro, 2018*). The functional modulation of VE-cadherin predominantly occurs through tyrosine phosphorylation and dephosphorylation of the VE-cadherin/catenin complex. Notably, vascular endothelial growth factor (VEGF) has the capacity to induce tyrosine phosphorylation of VE-cadherin, leading to enhanced endocytosis of VE-cadherin. Consequently, this downregulates the VE-cadherin's adhesion functions and ultimately increases the endothelial cell permeability (*Garnier & Vilgrain, 2023*).

DLL4 is highly expressed in vascular endothelium and is the exclusive Notch ligand within endothelial cells. It is a type I single transmembrane protein known for its exceptional specificity (*Shutter et al., 2000*). It is known to potentially inhibit tip cell

formation and govern the processes of vascular sprouting and vessel branching *via* Notch signaling (*Fournier et al., 2019*). While VE-cadherin and DLL4-Notch signaling play pivotal roles in angiogenesis and vascular function, their impact on AVF maturation remains unexplored.

Additionally, previous studies have presented a controversy regarding the role of intimal hyperplasia in AVF failure, with som studies denying its significant contribution (*Cheung et al., 2017*; *Vazquez-Padron et al., 2021*). These studies suggest that the remodeling of the vessel wall and the abnormal activation of vascular smooth muscle cells (VSMCs) may be equally important. Furthermore, the role of the vasa vasorum in the microenvironment of the vessel cannot be overlooked, as it may influence the maturation and long-term function of the fistula by modulating the local inflammatory environment and providing cellular signals (*Barcena et al., 2022*). We believe that this debate reflects the complexity of AVF maturation mechanisms and underscores the necessity for additional research to clarify the interplay among diverse factors.

This study aims to investigate the expression of VE-cadherin and DLL4 in the cephalic vein and serum during the initial AVF surgery in uremic patients and analyze their correlation with vascular maturation. We hypothesize that VE-cadherin and DLL4 may play a key role in the process of AVF maturation and could potentially serve as promising targets for future interventions.

## MATERIALS AND METHODS

### Study population

A total of 55 uremic patients who underwent their initial AVF surgery between August 2022 and May 2023 enrolled in our study and followed up for 3 months after surgery. The study was approved by the Medical Ethics Committee at the Affiliated Hospital of Xuzhou Medical University (Project Code XYFY2022-KL278) and registered in the Chinese Clinical Trial Registry (Registration number: ChiCTR2300074884). All patients enrolled were thoroughly briefed on the objectives of this research and provided their informed consent by signing the consent form. Inclusion criteria were patients diagnosed with uremia, patients undergoing hemodialysis as kidney replacement therapy and undergoing the first AVF surgery, the surgical method was the wrist cephalic vein-radial artery end-to-side anastomosis, and AVF matured without the use of any auxiliary techniques. Exclusion criteria were individuals who had previously undergone AVF surgery, were at risk of thrombosis due to preoperative arteriovenous thrombosis or hereditary thrombophilia, utilized a surgical approach different from radial artery lateral anastomosis, experienced severe infections, fever, or significant cardiovascular complications, had already started a routine hemodialysis program, had a central catheter, or had undergone urgent dialysis. In addition, smokers or patients receiving statin therapy were excluded. Previous studies have shown that cigarette smoking can increase the serum levels of vascular endothelial growth factor-A (VEGF-A), with a significant effect on AVF failure (*Haddad et al., 2019*; *Ozdemir et al., 2005*) while statins can improve the patency and maturation of AVF (*Cui et al., 2020*; *Janardhanan et al., 2013*).

## Sample size calculation

The sample size for this study was meticulously determined using PASS 15.0.5 software to ensure statistical rigor and sufficient power to detect significant outcomes at a predefined confidence level and anticipated effect size. The calculation aimed for a two-sided 95% confidence interval with a width of 0.2, employing the Clopper-Pearson exact method, requiring a sample size of 44 for the desired precision, assuming a 0.9 proportion of positive outcomes. To counter potential dropouts, the sample size was proactively increased by 20% to 55%, ensuring at least 44 evaluable participants for analysis. This adjustment is vital for maintaining the study's statistical power despite participant loss, thereby strengthening the study's statistical framework for precise and reliable parameter estimation (*Senn, 2011*; *Julious, 2005*).

## Research technique

### Clinical data

The clinical information of the patients, such as age, gender, primary ailment, date of surgery, blood pressure, as well as the diameters of the radial artery and cephalic vein, were gathered upon enrollment. Blood samples were taken for a full blood count, biochemical assessments, and flow cytometric analysis 24 h prior to the AVF procedure. Following three cases of thrombosis 1 week post-surgery, one patient passed away within 3 months, and seven individuals were no longer available for follow-up, resulting in a total of 44 patients being included in the study.

### Collection, preservation, and detection of the vascular specimens

The fistulas created in all patients were of radiocephalic type and underwent end-vein–to–side-artery anastomosis by the same surgeon. The most distal end of the cephalic vein was taken as a cephalic vein specimen, around 0.5 to 1 cm in length, and ensuring that the specimens did not interfere with the outcome of surgery. The residual blood from the cephalic vein sample was washed away with normal saline, and the specimen was immersed in 10% neutral formaldehyde (at a ratio exceeding 20:1), stored at 4 °C for at least 24 h, and later paraffin-embedded for staining. Standard procedures included HE, VE-cadherin, and DLL4 staining were performed subsequently.

### Methods of outcome determination

Hematoxylin-eosin (HE) staining: Sections were observed and photographed with a light microscope. We randomly selected five different positions from each slice, measured the thickness of the intima, and then we selected the average.

Immunohistochemical staining: Staining results were assessed using a light microscope, with a brown-yellow color change, indicating positive results. High-quality images were captured with a color pathological graphic analysis system. VE-cadherin and DLL4 expression were identified by optical density. The Fiji image analysis system determined the integrated optical density (IOD) within positively stained areas, and the average IOD was computed from four visual fields per section.

*Collection and testing of blood samples*

Blood specimens were collected in the morning following a 12-h overnight fast. The complete blood count was analyzed using the BECKMAN COULTER UniCel DxH 800, while biochemical parameters were assessed using the BECKMAN COULTER AU5800 (Beckman Coulter, Brea, CA, USA).

Fasting cubital venous blood samples of 3 ml were obtained from participants in the morning prior to surgery. Subsequently, the samples were centrifuged at 3,000 rpm for 10 min after being left at room temperature for 15 min, and the resulting supernatants were preserved at −80 °C in EP tubes. Human serum VE-cadherin and DLL4 levels were quantified utilizing an ELISA kit from MEIMIAN Company, adhering closely to the provided instructions.

## Evaluation of fistula maturation

Hemodialysis Fistula Maturation study considers that a mature fistula is one that can provide prescribed dialysis consistently with two needles for more than 75% of dialysis sessions within four consecutive weeks and maintain a blood flow rate of at least 300 ml/min (*Dember et al., 2014*). KDOQI guidelines suggest that AVF maturation should be based on clinical judgment (*Lok et al., 2020*). Before the first puncture of the arteriovenous fistula, we measured the diameters of the cephalic vein 5 cm proximal to the anastomosis and the blood flow velocity of the brachial artery 7 cm proximal to the elbow on the ipsilateral AVF (*Minxia et al., 2022*). Three months after surgery, a Doppler ultrasound was performed again to assess the maturity of the fistula. Based on the results, patients were classified into two groups: mature and failure. A comparison of different parameters was conducted between these groups.

## Statistical analysis

SPSS 26.0 statistical analysis software was utilized and GraphPad Prism 5.0 for data visualization. Normally distributed data are presented as mean ± standard deviation and were assessed with t-tests. Non-normally distributed data, described as median (M (P25, P75)), used non-parametric rank sum tests for comparisons. Categorical variables were expressed as percentages and compared using Fisher's exact test. Linear regression analyzed correlations between DLL4, VE-cadherin, and multiple variables, while binary Logistic regression assessed correlations between fistula immaturity and covariates. Statistical significance was set at ($P < 0.05$).

## RESULTS

The study population's demographic and clinical profiles are summarized in Table 1. Among the 44 study participants, 36 were male and eight were female, with an average age of 50 years. The follow-up period extended to 3 months, during which 30 cases progressed to mature fistula formation, while 14 cases resulted in failure fistulas. The mean time to AVF maturation was 55.86 ± 6.40 days. Before the first puncture of the arteriovenous fistula, the average diameter of the cephalic vein was 0.51 ± 0.08 cm and the mean blood

**Table 1 Demographic and clinical characteristics of the patients.**

| Parameters | Values ($n$ = 44) |
|---|---|
| Gender ($n$(%)) | |
| Male | 36 (81.8) |
| Female | 8 (18.2) |
| Age (years) | 50.36 ± 16.36 |
| Mean (±SD) | 19–79 |
| Rang | |
| BMI (kg/m$^2$) | 23.48 (20.62, 24.88) |
| Systolic pressure (mmHg) | 148.66 ± 16.07 |
| Diastolic pressure (mmHg) | 83.98 ± 16.75 |
| Fistula maturation ($n$(%)) | |
| Mature | 30 (68.2) |
| Failure | 14 (31.8) |
| CKD reason ($n$(%)) | |
| Diabetes mellitus | 12 (27.3) |
| Hypertension | 13 (29.5) |
| Glomerulonephritis | 8 (18.1) |
| Polycystic kidney disease | 4 (9.1) |
| Others | 7 (15.9) |
| Radial artery diameter (cm) | 0.23 ± 0.03 |
| Cephalic vein diameter (cm) | 0.24 ± 0.03 |
| Fistula maturation duration (days) | 55.86 ± 6.40 |
| Cephalic vein diameter at first puncture (cm) | 0.51 ± 0.08 |
| Blood flow at the brachial artery before the first puncture (mL/min) | 890 (569, 1,060) |

Notes:
Values are expressed as $n$ (%), mean ± SD or median (IQR).
BMI, body mass index; SD, standard deviation; CKD, chronic kidney disease; IQR, interquartile range.

flow at the brachial artery was 890 (569, 1,060) mL/min. Hypertension (13, 29.5%) and type 2 diabetes mellitus (T2DM) (12, 27.3%) were found to be the most common underlying factors leading to end-stage kidney disease (ESKD).

Our enrolled subjects were divided into two groups, the mature group 30, and the failure group 14 in terms of the presence of AVF maturation and were compared with each other. The mature group consisted of 26 men and four women, with a median BMI of 24.23 (22.51–24.98) kg/m$^2$ and a cephalic vein diameter of 0.25 (0.23–0.26) cm. The failure group comprised nine men and five women, with a BMI of 21.78 (20.01–23.40) kg/m$^2$ ($P$ = 0.049) and a cephalic vein diameter of 0.23 (0.21–0.24) cm, ($P$ = 0.005). The median VE-cadherin expression was 95.40 (79.03–107.16) ng/L in the failure group, and 125.07 (106.77–167.65) ng/L ($P$ = 0.001) in the mature group. The level of DLL4 expression was 92.78 (83.83–106.72) pg/mL in the mature group, and 60.42 (43.98–80.15) pg/mL ($P$ = 0.002) in the failure group. These differences in serum VE-cadherin and DLL4 levels between the two groups were statistically significant ($P$ < 0.05). There were no notable differences in radial artery diameter, hemoglobin, blood creatinine, BUN, serum calcium, blood phosphorus, and PTH levels between the two groups (Table 2).

**Table 2 Comparison of the parameters between mature and failure groups.**

| Parameters | Comparison of the groups | | P |
|---|---|---|---|
| | Mature group (*n* = 30) | Failure group (*n* = 14) | |
| Gender (*n*(%)) | | | 0.117 |
| Male | 26 (86.7) | 9 (64.3) | |
| Female | 4 (13.3) | 5 (35.7) | |
| Age (years, mean ± SD) | 56.63 ± 15.53 | 42.64 ± 15.03 | 0.08 |
| BMI (kg/m$^2$) | 24.23 (22.51, 24.98) | 21.78 (20.01, 23.40) | **0.049** |
| Systolic pressure (mmHg) | 151.43 ± 16.20 | 142.71 ± 14.57 | 0.094 |
| Diastolic pressure (mmHg) | 87.20 ± 17.20 | 77.07 ± 21.55 | 0.112 |
| Diabetes mellitus (*n*(%)) | | | 0.738 |
| Present | 11 (36.7) | 4 (28.6) | |
| Absent | 19 (63.3) | 10 (71.4) | |
| Radial artery diameter (cm) | 0.23 ± 0.02 | 0.22 ± 0.04 | 0.258 |
| Cephalic vein diameter (cm) | 0.25 (0.23, 0.26) | 0.23 (0.21, 0.24) | **0.005** |
| Fistula maturation duration (days) | 53.30 ± 4.68 | 61.35 ± 6.25 | **0.001** |
| Head vein diameter at first puncture (cm) | 0.55 ± 0.06 | 0.42 ± 0.04 | **0.001** |
| Blood flow at the brachial artery before the first puncture (mL/min) | 1,016 (885, 1,287) | 552 (526, 610) | **0.001** |
| Hemoglobin (g/L) | 91.50 (73.25, 101) | 91.50 (77.25, 99.5) | 0.811 |
| PTH (pg/mL) | 258.65 (146.9, 397.03) | 262.90 (124.33, 429.63) | 0.999 |
| Creatinine (umol/mL) | 692.50 (541, 893) | 657.50 (442.75, 1044) | 0.435 |
| BUN (mmol/L) | 26.84 (22.92, 34.28) | 21.00 (12.73, 34.12) | 0.112 |
| Calcium (mmol/L) | 2.11 (1.97, 2.18) | 2.05 (1.93, 2.09) | 0.140 |
| Phosphorus (mmol/L) | 1.82 (1.55, 2.26) | 1.72 (1.37, 2.51) | 0.950 |
| DLL4 (pg/mL) | 92.78 (83.83, 106.72) | 60.42 (43.98, 80.15) | **0.002** |
| VE-cadherin (ng/L) | 125.07 (106.77, 167.65) | 95.40 (79.03, 107.16) | **0.001** |

**Notes:**
Values are expressed as *n* (%), mean ± SD or median (IQR).
BMI, body mass index; SD, standard deviation; PTH, parathyroid hormone; BUN, blood urea nitrogen; IQR, interquartile range.
Bold indicates statistical significance ($P < 0.05$).

By employing linear regression analysis with DLL4 and VE-cadherin content as the dependent variables, and considering predictive variables such as sex, age, BMI, presence of T2DM, Systolic pressure, radial artery diameter, and head artery diameter, the findings indicated a highly significant positive correlation between DLL4 expression levels and VE-cadherin expression levels ($P < 0.001$) (Table 3).

The binary logistic regression analysis revealed a significant association between the presence of fistula immaturity and cephalic vein diameter, VE-cadherin, and DLL4 expression levels were statistically significant ($P < 0.05$). Even after accounting for potentially confounding variables such as gender, age, BMI, diabetes history, and systolic blood pressure, VE-cadherin, and DLL4 maintained a significant correlation with fistula immaturity (Table 4). Comparison of the receiver operating characteristic (ROC) curves was made between the VE-cadherin, DLL4, and cephalic vein diameter variables, the curves were 0.919, 0.912 and 0.898, respectively (Fig. 1). These findings indicate that VE-cadherin and DLL4 expression levels may be related to the occurrence of immature

**Table 3 Linear correlation analysis between DLL4 and VE-cadherin expression levels and multivariate variables.**

| Variables | DLL4 (pg/mL) | | VE-cadherin (ng/L) | |
|---|---|---|---|---|
| | Pearson/spearman | | Pearson/spearman | |
| | r | P | r | P |
| Gender | −0.22 | 0.886 | −0.118 | 0.447 |
| Age | 0.217 | 0.157 | 0.282 | 0.063 |
| BMI | 0.001 | 0.993 | 0.146 | 0.343 |
| Diabetes mellitus | −0.057 | 0.715 | −0.093 | 0.550 |
| Systolic pressure | 0.016 | 0.920 | 0.015 | 0.921 |
| Radial artery diameter | 0.124 | 0.423 | −0.097 | 0.53 |
| Cephalic vein diameter | −0.028 | 0.855 | −0.024 | 0.877 |
| VE-cadherin (ng/L) | 0.784** | **<0.001** | – | – |
| DLL4(pg/mL) | – | – | 0.784** | **<0.001** |

Notes:
BMI, body mass index.
Bold indicates a correlation between the two variables ($P < 0.05$).
** A strong correlation.

**Table 4 Correlation analysis between fistula immaturity and multivariate variables.**

| Models and variables | OR [95%CI] | P |
|---|---|---|
| Raw model | | |
| Cephalic vein diameter (cm) | 0.970 [0.922–1.021] | **0.01** |
| VE-cadherin (ng/L) | 0.960 [0.930–0.992] | **0.014** |
| DLL4 (pg/mL) | 0.966 [0.939–0.994] | **0.018** |
| Adjust the confounding factor model (including sex, age, BMI, history of diabetes, and systolic blood pressure) | | |
| Cephalic vein diameter (cm) | 0.930 [0.874–0.990] | **0.036** |
| VE-cadherin | 0.961 [0.929–0.995] | **0.024** |
| DLL4 (pg/mL) | 0.958 [0.926–0.992] | **0.014** |

Notes:
OR, odds ratio; CI, confidence interval.
Bold values are statistically significant ($P < 0.05$).

fistulas and hold promise as indicators for predicting the maturation status of internal fistulas.

## Analysis of the HE staining

A total of 14 patients underwent HE staining of cephalic vein tissue, with 8 subjects in the mature group and six subjects in the failure group (Fig. 2). We conducted measurement of venous intima and media thickness in both groups, followed by a comparison of them. The results indicated that the vascular intima and media thickness in the failure group surpassed that in the mature group, albeit without reaching statistical significance ($P > 0.05$) (Figs. 2C and 2D).

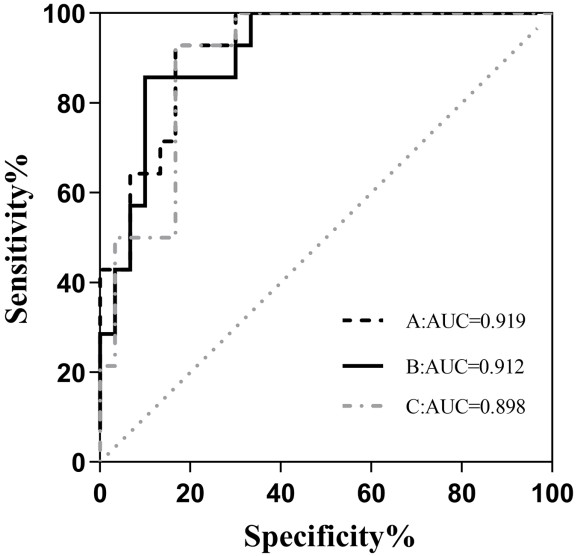

A:Log(VE-cadherin);B:Log(DLL4);C:Log(Cephalic vein diameter)

**Figure 1 Comparison of the ROC curves between VE-cadherin, DLL4, and cephalic vein diameter variables as predictors of fistula maturation.** AUC, Area under curve A, B, and C correspond to the predictive probability values of each predictor variable, derived from binary logistic regression analysis. This analysis incorporates sex, age, BMI, history of diabetes, and systolic blood pressure as confounding variables, while considering VE-cadherin, DLL4, and cephalic vein diameter as predictive variables.

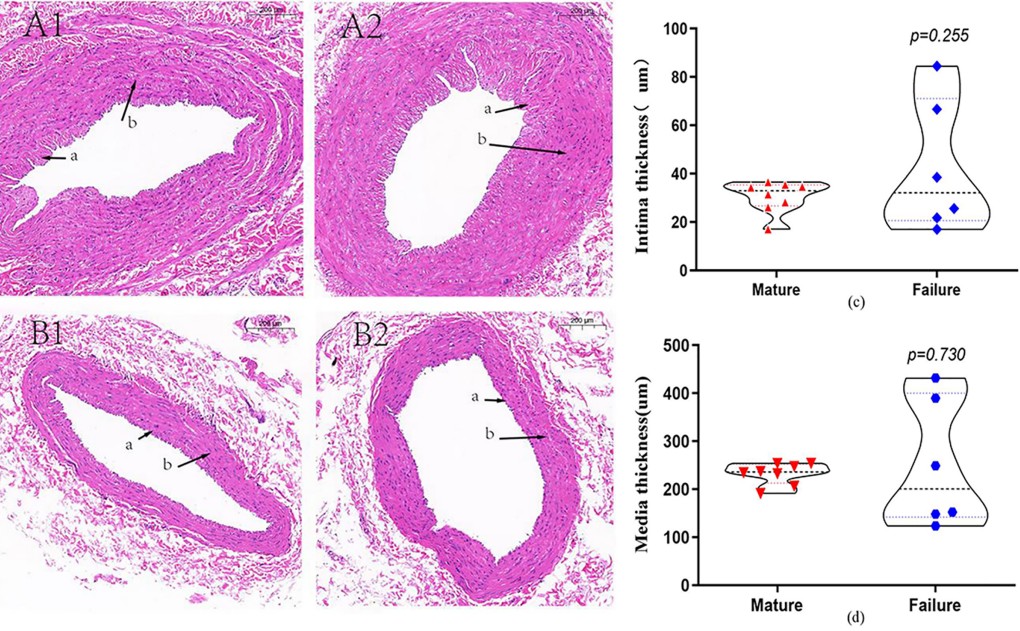

**Figure 2 HE staining in the mature group (A) and the failure group (B) (5×).** Among these patients, more than 70% exhibited distinctive histological features, including endothelial injury, endothelial cell hyperplasia, subendothelial fibrous tissue hyperplasia, extracellular matrix accumulation, and smooth muscle cell hyperplasia, resulting in eccentric lumen stenosis (A). Moreover, the media layer displayed extensive smooth muscle cell proliferation, along with hyaline degeneration and infiltration of inflammatory cells (B). Photo credit: Huanhuan Yin.

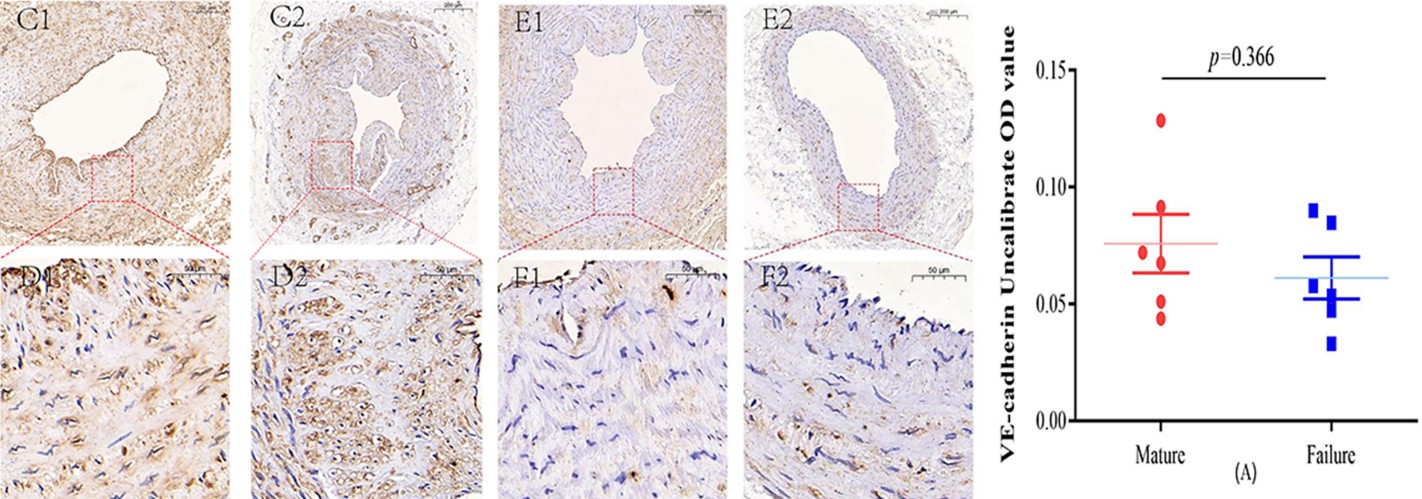

**Figure 3 Immunohistochemical staining of VE-cadherin mature group (C/D) and failure group (E/F) (5×, 20×).** VE-cadherin predominantly manifests its expression within the endothelial layer. While VE-cadherin expression levels were comparatively higher in the mature group than in the failure group, this disparity did not attain statistical significance ($P = 0.366$) (A). Photo credit: Huanhuan Yin.

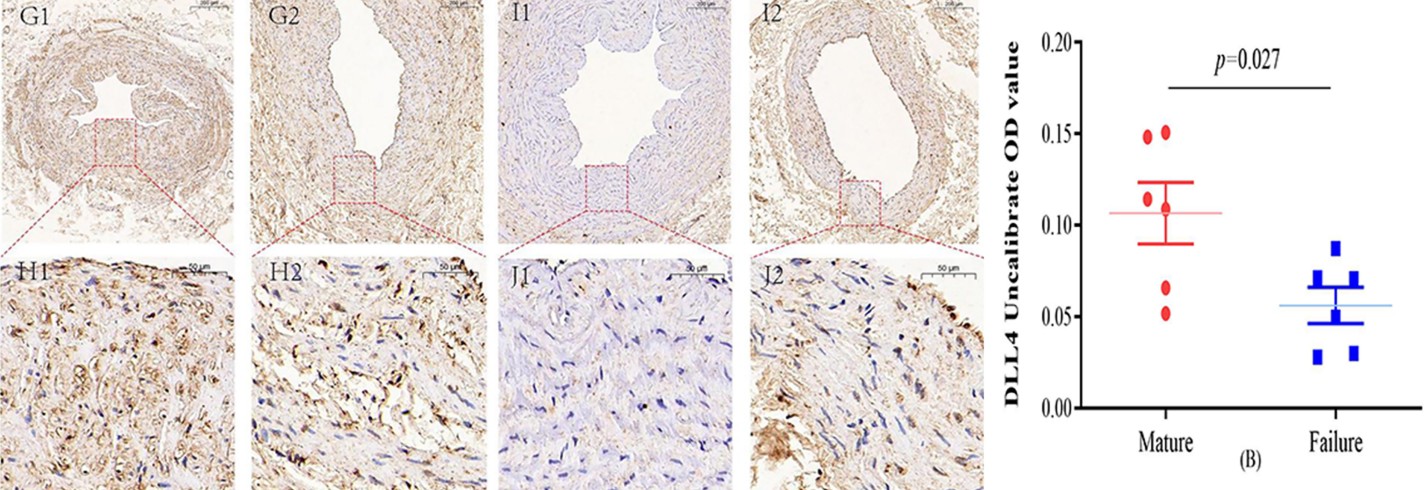

**Figure 4 Immunohistochemical staining of DLL4 mature group (G/H) and failure group (I/J) (5×, 20×).** In the mature group, DLL4 was predominantly localized in the cell membrane and cytoplasm, primarily within the inner membrane. In contrast, DLL4 expression was notably weaker in the failure group. However, the DLL4 expression level in the mature group significantly exceeded that in the failure group, displaying a noteworthy statistical difference ($P = 0.027$) (B). Photo credit: Huanhuan Yin.

## Immunohistochemical staining

Immunohistochemical staining was performed to observe the expression of VE-cadherin and DLL4 in the vascular endothelium, we found that both VE-cadherin and DLL4 were clearly expressed in the endothelial layer, but the mature group was higher than that in the failure group. The results of the two groups are shown in Figs. 3 and 4.

## DISCUSSION

The current study investigated the interplay of VE-cadherin and DLL4 expressions within the cephalic vein of uremic patients as well as their serum levels, shedding light on the potential relationship with AVF maturation. Our findings indicated that VE-cadherin and DLL4 were predominantly expressed in the endothelial cell layer. Notably, DLL4 expression was significantly higher in the mature group, suggesting that reduced DLL4 expression in the local vascular environment may contribute to AVF dysfunction in uremic patients. This could be linked to endothelial cell differentiation, migration, and proliferation processes. Conversely, VE-cadherin expression within the vascular wall did not significantly decrease in patients with AVF maturation failure, highlighting the need for further investigation into the role of these markers in vascular remodeling. The study also revealed that serum levels of VE-cadherin and DLL4 were higher in the mature group, with a significant correlation between these levels and fistula maturity. This implies that serum expression of these markers may hold relevance to fistula occurrence and could offer valuable insights for predicting fistula maturity.

AVF maturation is a complex process involving the venous endothelium and adventitia (*Gasper et al., 2013*), with various biological factors influencing its success, including inflammation, vascular smooth muscle cell proliferation, matrix turnover, and endothelial cell damage (*Shiu et al., 2019*). *Zhao et al. (2017)* found that vascular smooth muscle cells in a differentiated state can promote the development of neointimal hyperplasia (NIH), which is a primary contributor to luminal narrowing and eventual AVF malfunction. The intricate process of NIH triggering AVF failure involves various factors, including oxidative stress, nitric oxide, and the MMP family and damage to endothelial cells (*Gasparin et al., 2022*). In addition, the vascular microenvironment influences the functionality and maturation of arteriovenous fistulae (AVF) through a combination of factors including hemodynamic changes, inflammatory responses, and hypoxia (*Barcena et al., 2022*). *Rai & Agrawal (2022)* have specifically focused on the changes in the phenotype of vascular smooth muscle cells (VSMCs) within a microenvironment enriched by increased immune cell infiltration and cytokine secretion. Their research indicates that impaired immune cell infiltration and alterations in the phenotype of VSMCs play a critical role in the pathophysiology of failed maturation of AVFs.

Oxidative stress plays a significant role at every stage of AVF maturation, including preoperative underlying mechanisms, intraoperative surgical damage, and postoperative hemodynamic alterations. A growing body of evidence suggests a link between AVF failure and oxidative stress (*Hu et al., 2023*). Endodermal nitric oxide (NO), produced by endothelial nitric oxide synthase (eNOS), plays a crucial role as a vasodilator and signaling molecule in vascular remodeling, potentially facilitating adaptive venous wall remodeling through its anti-inflammatory, antithrombotic, and antiproliferative characteristics. Another study showed that eNOS can inhibit neointimal hyperplasia in AVF. Increased blood and luminal flow induce eNOS activation while inhibiting the expression of eNOS can impair endothelial function (*Bai et al., 2021*). *Sadaghianloo et al. (2019)* found that both eNOS and inducible nitric oxide synthase (iNOS) were upregulated in the AVF.

*Somarathna et al. (2022)* found that the NOS3-NO-cGMP pathway is positively associated with reduced AVF venous intimal hyperplasia, and that upregulated NOS3 expression can promote desired AVF remodeling. This was also demonstrated by *Pike et al. (2019)*. In AVFs, the MMP family is the main factor regulating extracellular matrix degradation, with MMP-2 and MMP-9 upregulated in the early stage; an increase in serum MMP-2 levels serves as an important indicator for predicting AVF maturation (*Tronc et al., 2015*). In addition, TGF-β signaling has been shown to play a central role in vascular remodeling of AVFs, and TGF-β has been proposed to be closely related to intimal hyperplasia (*Xie et al., 2022*). Reactive oxygen species (ROS) can induce extracellular matrix (ECM) remodeling and fibrosis by activating the TGF-β1/Smad3 pathway, upregulating connective tissue growth factor, and increasing MMP-2 expression. Additionally, ROS can also enhance TGF-β1 levels by activating the MAPK/ERK signaling pathway, promote MMP-9 expression and secretion in vascular smooth muscle cells (VSMCs), and regulate ECM metabolism (*Hu et al., 2022*).

Vascular endothelial growth factor (VEGF) can exert its effects selectively on vascular endothelial cells, orchestrating a range of vital functions such as stimulating endothelial cell mitosis, promoting endothelial cell survival, and regulating vascular permeability (*Malyszko, 2010*). VEGF receptors encompass VEGFR-1 and VEGFR-2. VEGFR-1 plays a pivotal role in endothelial cell differentiation and migration, while the activation of VEGFR-2 propels endothelial cells into a state of angiogenesis, instigating their migration and proliferation (*Melincovici et al., 2018*). VEGF-A emerges as a central player during angiogenesis. It has the ability to kick-start cell proliferation, inhibit cell apoptosis, heighten vascular permeability, facilitate vascular dilation, and recruit inflammatory cells to the injury site. In recent studies, several mechanosensory, such as integrins, platelet endothelial cell adhesion molecule-1 (PECAM-1)/vascular endothelial cell (VE-cadherin)/ vascular endothelial growth factor receptor 2 (VEGFR2) complexes, and notch1, have been shown to contribute to mechanotransduction and vessel remodeling (*Wen et al., 2023*).

During the formation of an AVF, there may be disturbances in local blood flow and increased pressure on the vascular walls. In this situation, local tissues may release VEGF to promote the formation of new blood vessels in response to the increased pressure from the augmented blood flow. *Chen et al. (2022)* found that in failed AVFs, the vascularization of the vessel wall, particularly in the regions of neointimal hyperplasia (NIH), was significantly more developed. This increased vascularization may be associated with the high expression of vascular endothelial growth factor A (VEGF-A), which could promote neovascularization and vascular remodeling. *Huang et al. (2021)* also demonstrated that down-regulating the expression of the VEGF-A gene can inhibit the formation of intimal hyperplasia during AVF maturation.

VE-cadherin, a cadherin exclusive to vascular endothelial cells, is particularly crucial in vascular development and remodeling, actively participating in guiding the differentiation, proliferation, and migration of endothelial cells, thereby promoting the formation of new blood vessels and triggering vascular wall thickening (*Claesson-Welsh, Dejana & McDonald, 2021*). However, when VE-cadherin undergoes phosphorylation, it sets in motion a cascade of events. This includes heightened vascular permeability and the

internalization of VE-cadherin, culminating in the breakdown of the vascular barrier (*Gavard, 2014*). VEGF orchestrates can elevate vascular permeability and endothelial migration during angiogenesis by finely tuning the functioning of VE-cadherin (*Ninchoji et al., 2021*). VEGFR-2 collaborates closely with VE-cadherin in upholding the integrity of endothelial cell barriers. In the presence of VEGF, this growth factor binds to VEGFR-2, activating and internalizing the receptor, thus triggering a signaling cascade. This cascade, in turn, prompts the internalization and subsequent degradation of VE-cadherin resulting in the disruption of adherens junctions and a consequential increase in vascular endothelial permeability (*Zhou et al., 2022*). In addition, studies have found that the incidence rate of antiphospholipid antibodies (APL) in hemodialysis (HD) patients was higher than that in the general population or conservative treatment of ESKD patients (up to 37%), which seemed to be related to the failure of AVF maturation, possibly caused by the impairment of vascular remodeling ability caused by endothelial dysfunction (*Taghavi et al., 2024*). *Simone et al. (2002)* found that APL can downregulate the level of VE-cadherin in nourishing cells.

DLL4 is a transmembrane ligand that plays a pivotal role in the Notch receptor signaling pathway. It is involved in angiogenesis, negatively regulating endothelial cell proliferation and migration, and contributes to vascular development (*Meng et al., 2016*). In normal vascular development, DLL4 expression is tightly regulated and serves as an inhibitory regulator during neovascularization. When DLL4, located within vascular endothelial cells, binds to the Notch receptor, it can suppress the differentiation of nearby endothelial cells. This inhibitory role is crucial for maintaining the balance of vascular endothelial cell homeostasis and ensuring the formation of normal vascular structures. In a study by *Guo et al. (2022)* they unveiled that activated Notch signaling not only propelled the activation and differentiation of vascular smooth muscle cells but also wielded regulatory control over vascular remodeling in AVF and neointima formation. Deficiencies in Notch signaling can result in impaired vascular smooth muscle cells, provoking endothelial cell (EC) barrier dysfunction and inflammation, ultimately culminating in the failure of AVF maturation. Notch and VEGF signaling work in tandem to direct the formation of tip and stalk cells within the endothelial layer (*Pulkkinen et al., 2021*). Studies have also demonstrated that VEGF significantly upregulates the expression of DLL4 mRNA (*Best, Moran & Ren, 2018*). DLL4/Notch signaling operates downstream of VEGF signaling and plays a crucial role in the regulation of vascular remodeling during embryonic vascular development (*De Zoysa et al., 2022*). *Trindade et al. (2017)* used a transgenic mouse model with endothelial cell-specific DLL4 overexpression. They found that increasing DLL4/Notch signaling could reduce VEGF-induced endothelial cell proliferation. Conversely, inhibiting DLL4/Notch signaling led to lumen morphology defects and dysfunctional parietal cell accumulation, impeding functional blood vessel formation. However, some studies report that Notch in endothelial cells of AVF might lead to neointimal hyperplasia and, therefore, AVF failure. *Wang et al. (2014)* demonstrated that Notch signaling is a crucial activator in the pathways regulating endothelial function and neointimal formation. Elevated TGF-β1 may activate Notch within the endothelial cells of AVF, resulting in expedited neointimal formation and AVF failure. Suppressing Notch

activation could potentially improve the function of the AVF in patients suffering from uremia. Therefore, the role played by DLL4/Notch in AVF maturation deserves further exploration.

In addition, our study revealed that compared to the mature group, the failure group had significantly smaller cephalic vein diameters, and there was a clear correlation between vein diameter and fistula immaturity. These findings align with the previous findings in a study conducted by *Bashar et al. (2016)*. Previous results have indicated that both vein and artery diameters significantly impact AVF maturity, with vein diameter possibly serving as an independent predictive indicator for AVF maturity (*Bahi, 2022*).

The pathogenesis of arteriovenous fistula (AVF) failure in uremic patients is multifactorial, with intimal hyperplasia (IH) traditionally implicated as a key contributor. Our study, aligning with recent literature, challenges this paradigm by demonstrating no significant association between preoperative intimal hyperplasia and AVF maturation failure. *Oprea et al. (2017)* reported that veins with histopathological alterations before surgery, such as intima hyperplasia, may not exert a decisive influence on AVF maturation at the time of fistula formation. *Tabbara et al. (2016)* also proved that pre-existing, postoperative, and change in IH over time were not associated with AVF outcomes. Despite the prevalence of IH in the vasculature of uremic patients, its clinical impact appears to be modulated by a complex interplay of factors beyond mere thickness.

Despite the valuable insights gleaned from this study regarding VE-cadherin and DLL4 in vascular maturation post-AVF, the limitations of our study are acknowledged. Firstly, the sample size, while adequate for our analysis, may not have provided sufficient power to detect subtler associations between IH and AVF failure. Secondly, we solely assessed the expression levels of VE-cadherin in established AVF venous endothelial cells without delving into its expression during hyperplasia proliferation throughout AVF maturation. This narrow focus may have implications for the overall findings. Lastly, there could be unidentified variables at play, such as the potential of drug interventions, capable of influencing the expression of these markers within the vascular endothelium.

Future research directions should aim to elucidate the intricate interplay between the vasa vasorum, the endothelium, and the microenvironment in the context of AVF maturation. Longitudinal studies with larger cohorts and the incorporation of advanced imaging techniques to assess the vasa vasorum could provide deeper insights. Additionally, exploring the molecular signatures of endothelial cells within the AVF may uncover novel biomarkers or therapeutic targets.

## CONCLUSIONS

In conclusion, VE-cadherin and DLL4 bear substantial physiological and pathological significance in endothelial cell proliferation, migration, and vascular remodeling. Subsequent studies may unveil their specific roles in AVF maturation, pinpoint fresh avenues for preventing AVF failure, and lay the groundwork for developing therapeutic strategies.

## ACKNOWLEDGEMENTS

We are grateful to our colleagues and laboratory for providing the care and support for conducting our research.

### Funding

This study was supported by funding from the National Natural Science Foundation of China (82270731), the Jiangsu Provincial Natural Science Foundation (BK20211054), the Jiangsu Provincial Commission of Health and Family Planning (2016103003, H201628), a project of Jiangsu Provincial Post Graduate Innovation Plan (KYCX17_1708, SJCX17_0560, KYCX18-2178, SJCX18_0715), the Science and Technology Development Fund of Affiliated Hospital of Xuzhou Medical University (XYFC2020001), the Xuzhou Medical leading Talent training Project (XWRCHT20210038), the Xuzhou key R & D Program (Social Development) (KC20160), the New Technology project of Affiliated Hospital of Xuzhou Medical University (2020301018) and the key research and development plan (social development) project of Xuzhou city (KC23332). The funders had no role in study design, data collection and analysis, decision to publish, or preparation of the manuscript.

### Grant Disclosures

The following grant information was disclosed by the authors:
National Natural Science Foundation of China: 82270731.
The Jiangsu Provincial Natural Science Foundation: BK20211054.
The Jiangsu Provincial Commission of Health and Family Planning: 2016103003, H201628.
Jiangsu Provincial Post Graduate Innovation Plan: KYCX17_1708, SJCX17_0560, KYCX18-2178, SJCX18_0715.
Affiliated Hospital of Xuzhou Medical University: XYFC2020001.
Xuzhou Medical leading Talent training Project: XWRCHT20210038.
Xuzhou key R & D Program (Social Development): KC20160.
Affiliated Hospital of Xuzhou Medical University: 2020301018.
Key Research and Development Plan (Social Development): KC23332.

### Competing Interests

The authors declare that they have no competing interests.

### Author Contributions

- Huanhuan Yin conceived and designed the experiments, performed the experiments, analyzed the data, prepared figures and/or tables, authored or reviewed drafts of the article, and approved the final draft.
- Yifan Tang performed the experiments, authored or reviewed drafts of the article, and approved the final draft.

- Yanping Wang performed the experiments, analyzed the data, prepared figures and/or tables, and approved the final draft.
- Yousuf Abdulkarim Waheed analyzed the data, authored or reviewed drafts of the article, and approved the final draft.
- Disheng Wang performed the experiments, authored or reviewed drafts of the article, and approved the final draft.
- Dong Sun conceived and designed the experiments, analyzed the data, authored or reviewed drafts of the article, and approved the final draft.

### Human Ethics

The following information was supplied relating to ethical approvals (*i.e.*, approving body and any reference numbers):

The Medical Ethics Committee at the Affiliated Hospital of Xuzhou Medical University approval to carry out the study within its facilities (Project Code XYFY2022-KL278)

### Data Availability

The raw measurements are available in the Supplemental Files.

### Clinical Trial Registration

The following information was supplied regarding Clinical Trial registration:

ChiCTR2300074884

### Supplemental Information

Supplemental information for this article can be found online at http://dx.doi.org/10.7717/peerj.18356#supplemental-information.

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
