# Peer review of "Correlation between pre-operative VE-cadherin and DLL4 and the maturation after primary arteriovenous fistula in uremic patients"

_PeerJ, doi:10.7717/peerj.18356_

## Round 0.1 · original submission · Major Revisions

Besides the reviewers suggestions, some other changes should be done:

- Please describe in more detail the selection process of the subjects.
- Please add comments on the role of oxidative stress and nitric oxide on the endothelial dysfunction and its potential interactions with arterial function and fistula failure.

·

Basic reporting

The study intended to demonstrate a correlation between endothelial markers and arteriovenous fistula maturation in a small cohort of patients undergoing forearm fistula creation.

Experimental design

The overall study is underpowered, single-center, and sex unbalanced – mostly males were included.

Validity of the findings

The future validation of the study is dubious.

Additional comments

The premises of the study were extrapolated from arterial biology. Most of the introduction is supported by opinion papers (reviews). For instance, Cheung et al. denied the actual contribution of juxtanastomotic anastomosis to fistula failure as part of the studies from the HFM, which are further cited in the discussion. The contribution of intimal hyperplasia to fistula failure has been denied in several studies (Tabbara et al. and Martinez et al.), which were ignored by the authors. The study's premises consider only the luminal EC a significant player in venous arterialization. However, most EC in the human vein resides in an extensive network of microvessels forming the vasa vasaruum. The inclusion and exclusion criteria are unclear.

Reviewer 2 ·

Basic reporting

This is a well-written research article exploring the pathophysiology of arteriovenous fistula (AVF) maturation failure, which is crucial for improving AVF patency rates and patient outcomes.

I suggest the authors read Haidi Hu et al., Semin Vasc Surg, 2016. This article could be used in place of or in addition to reference no. 6, which is an editorial.

Line 194: Please change Table 3 to Table 4.

Experimental design

Well conducted.Methods:

Only preemptive AVFs were included, with a surprisingly high AVF maturation rate compared to the literature, possibly due to the younger age, higher male rate, and lower diabetes prevalence in the cohort.
a. Can you provide the prevalence of smokers, cardiovascular disease (such as peripheral vascular disease and cerebrovascular disease), and treatments (statins), which are known risk factors for AVF maturation failure?
b. The methodology implies a follow-up of three months after AVF creation. Does this mean that all patients started hemodialysis (HD) at the same time (i.e. at 3 months)? Please provide the time from AVF creation to the first puncture as we know that maturation process can take time.
c. Is there a US evaluation before the first AVF puncture? If so, it would be very interesting to have the efferent vein diameter and blood flow. The presence of AVF stenosis during the maturation process is also very interesting.

Validity of the findings

Lines 214-215 : "The expression of VE-cadherin and DLL4 in the mature group was significantly increased compared to the failure group". For what i understand, only the expression of DLL4 in significantly higher.
Please discuss the hypothetical reasons why VE-cadherin is not lower in the vascular wall of patients with AVF maturation failure. Is this only related to the small sample size?

Lines 279-281: The authors state, "Deficiencies in Notch signaling can result in impaired vascular smooth muscle cells, provoking endothelial cell (EC) barrier dysfunction and inflammation, ultimately culminating in the failure of AVF maturation." However, some studies report that Notch in endothelial cells of AVF might lead to neointimal hyperplasia and, therefore, AVF failure. Blocking Notch in endothelial cells might prevent AVF failure in the setting of uremia. This phenomenon in uremic patients might be associated with the increase of TGF-β1 (Wang Y, et al. J Am Soc Nephrol. 2014, doi: 10.1681/ASN.2013050490). Please discuss this aspect of Notch regulation.

Lines 215-217: "This finding suggests that reduced expression of VE-cadherin and DLL4 in the local vascular environment may be related to the dysfunction of arteriovenous fistulas in patients with uremia."
It would be intersting to evaluate in another study fators that might be implicated in VE-cadherin downregulation and maturation failure. For example, antiphospholipid antibodies, which are prevalent in HD patients (up to 37%) and seems to be associated with AVF maturation failure, are known to down regulate VE-cadherin in trophoblast cells (Di Simone, N.; Castellani, R.; Caliandro, D.; Caruso, A. Antiphospholipid Antibodies Regulate the Expression of Trophoblast Cell Adhesion Molecules. Fertil. Steril. 2002, 77, 805–811).

---

## Round 0.2 · Major Revisions

Comments on the statistical power of the study need to be added to the methodology section.

·

Basic reporting

The reference remains ambiguous, primarily relying on opinion papers (reviews) and information from arterial biology to support potential associations. The authors did not address the questions on this matter and disregarded my previous comments.

Experimental design

The study remains underpowered, and the authors did not address my comments on this issue

Validity of the findings

The findings are not robust or statistically sounded.

Reviewer 2 ·

Basic reporting

No comment

Experimental design

The authors have responded well to the comments provided.

Validity of the findings

The authors have responded well to the reviews and comments provided.

Table 2. Make the p-value of BMI in Table 2 bold, as the value is significant.

Additional comments

Line 313 of the clean manuscript, the authors have switched references 41 and 42. "Di Simone, et al. found that APL can downregulate the level of VE-cadherin in nourishing cells"

---

## Round 0.3 · accepted · Accept

The previous suggestions were adequately resolved. No further comments.